# Particulate Cell Wall Materials of *Lactobacillus acidophilus* as Vaccine Adjuvant

**DOI:** 10.3390/vetsci9120698

**Published:** 2022-12-15

**Authors:** Shu-Ching Lin, Pu-Chieh Chang, Chien-Hung Lin, Hong-Jen Liang, Chih-Hung Huang

**Affiliations:** 1Department of Chemical Engineering & Biotechnology, Institute of Chemical Engineering, National Taipei University of Technology, Taipei 10608, Taiwan; 2Country Best Biotech Co., Ltd., Taipei 100411, Taiwan; 3Department of Food Science, Yuanpei University, Hsinchu 30015, Taiwan

**Keywords:** peptidoglycan, immunization, chicken, antibody titer, particle

## Abstract

**Simple Summary:**

Different materials such as aluminum salts and bacterial peptidoglycan are considered immunogenic enhancers. *Lactobacillus acidophilus* (LA), a Gram-positive bacterium also known as a probiotic, is considered safe, acts as an immune enhancer when purified to form a pure peptidoglycan particle and is a possible candidate for a vaccine adjuvant. LA particles (LAPs) treated with high-pressure homogenization (HPH) with the addition of trehalose and emulsifiers had an average diameter of 179 nm. A five-fold dosage of LAPs treated with HPH and additives can induce a higher antibody titer response compared with commercial adjuvants in murine species. In comparison with ISA70, LAPs can stimulate an even antibody titer response but this decreased more quicky after a few weeks in chickens. Different formulation combinations of carbomer and LAPs induce a similar antibody response to commercial ISA70 with no acute toxicity, suggesting that LAPs are a potent vaccine adjuvant.

**Abstract:**

We evaluated *Lactobacillus acidophilus* (LA) for adjuvant application in animal vaccines. LA particles (LAPs) are made by treating LA with purification processes and high-pressure homogenization (HPH). We found that LAPs treated with HPH with trehalose and emulsifiers had an average particle size of 179 nm, considerably smaller than LAPs without additives. First, we evaluated the adjuvanticity of LAPs using a murine model with ovalbumin antigens, revealing that LAPs, especially in a five-fold concentration, could induce a considerable antibody response compared with other current adjuvants. In poultry vaccination tests using inactivated Newcastle disease virus, LAPs alone could induce a similar antibody response compared to commercial water-in-oil (W/O) adjuvant ISA70, a commercial adjuvant, at weeks 4 and 6; however, they declined faster than ISA70 at weeks 8 and 10. LAPs added to conventional adjuvant materials, such as mineral oil-based O/W emulsions, showed similar adjuvanticity to ISA70. LA-H5-C, composed of carbomer, emulsifiers and trehalose showed no significant body weight change in acute toxicity compared to other adjuvants including ISA70, making formulated LAPs a potential candidate for use as a veterinary vaccine adjuvant.

## 1. Introduction

Currently, most vaccines are used for prophylactic purposes to prevent diseases such as influenza and, recently, COVID-19 [1]. In general, vaccines may contain live-attenuated or inactivated pathogens or merely segments of the pathogen (subunit vaccines) [2]. Live-attenuated vaccines can elicit long-term immunity; however, they have the risk of reverting into a new virulent strain [3]. Inactivated or subunit vaccines are considerably safer and possess higher stability, facilitating storage and transport [4]. However, these vaccines are relatively non-immunogenic and need an adjuvant to enhance their immunogenicity [5]. Adjuvants are generally divided into two main groups: (1) immunomodulatory molecules, which are mainly from pathogens that can activate innate immune receptors, such as Toll-like receptors (TLRs), and (2) delivery system formulations that act as carriers, adsorbing or entrapping antigens for protection or slow release, such as aluminum salts (alum), oil-in-water and water-in-oil emulsions, and liposomes [6]. 

Of all of the licensed vaccines, alum is the most widely used in both human and animal vaccines. Although alum has been proven safe and effective, its limitations are that it poorly induces cell-mediated immunity (CMI) [7]. CMI plays a vital role in protection against viral-infected cells [8]. T helper 1 (Th1) cells can induce CMI by activating B cells through secreting interferon-gamma to produce immunoglobulin (Ig) G1 and IgG3 in humans (IgG2a and IgG2b in mice) [9]. While subunit vaccines primarily activate T helper 2 (Th2) cells, combinations of subunit vaccine antigens with specific adjuvants can trigger Th1 to induce robust CMI [10]. Biomaterials such as cholera toxin subunit B, saponin extract QS-21, synthetic CpG oligodeoxynucleotides and bacterial flagellin are used to induce robust CMI [11].

*Lactobacillus acidophilus* (LA) is a Gram-positive bacteria that grows at low pH conditions (below pH 5.0) and is found in human and animal gastrointestinal tracts [12]. *L. acidophilus* is considered a probiotic, is supplemented in many dairy products and is known for improving and restoring gut flora [13]. Many applications regarding this bacterium have been found, including lowering serum cholesterol [14], preventing yeast infections, balancing vaginal microbiota [15] and treating pediatric diarrhea [16]. The Gram-positive enhancer matrix (GEM) made from *Lactobacillus lactis* can stimulate the host’s immune system by acting as an immune enhancer by its peptidoglycan component and act as carriers to adsorb antigens [17]. Through heat-inactivation and acidic treatment, bacterial debris composed of protein and DNA can be eliminated to form pure peptidoglycan particles that enhance immunity [18].

Here, we used LA followed by the process of GEM preparation with modified formula through disruption using a high-pressured homogenizer. The dispersed LA particles are 510 and 175 nm in size with no precipitation when trehalose and emulsifiers are added. To develop the peptidoglycan particle adjuvant, we first examined the adjuvants’ potential toxic effects and immunogenicity with ovalbumin (OVA) in murine models through intramuscular injection (IM) and their potential application for Newcastle disease (ND) vaccine in poultry models by IM. We also evaluated the potential application of LA particles as immune enhancers in conjunction with conventional adjuvants, such as polyacrylic acid polymer or oil emulsions.

## 2. Materials and Methods

### 2.1. Preparation of Lactobacillus acidophilus (LA) Particles

LA (BCRC 17049) was bought from Bioresource Collection and Research Center, Hsinchu City, Taiwan and stored in 20% glycerol at −80 °C (Model 996; Thermo, Waltham, US) until needed. LA was unfrozen and staked onto a MRS (1% proteose peptone, 1% beef extract, 0.5% yeast extract, 2% dextrose, 0.1% polysorbate 80, 0.2% ammonium citrate, 0.5% sodium acetate, 0.01% magnesium sulfate, 0.0005% manganese sulfate, and 0.2% dipotassium phosphate; STBIO MEDIA, Taipei, Taiwan) agar plate and incubated at 30 °C for 5 days. Three single colony were picked and inoculated into 3 separate 1L-glass-flasks filled with 300 mL of MRS broth each. The flasks were incubated at 30 °C for 2 days in an orbital shaker (S303R; FIRSTEK, Taipei, Taiwan) at 150 rpm. After 2 days, the cultured broth from the flasks were inoculated into a 10L-fermenter (FS-07; Major Science, Taipei, Taiwan) composed of 10 L MRS broth and incubated at 30 °C for 7 days. 

The harvested LA culture was washed twice with sterile distilled water. We resuspended cells in 0.1 M hydrochloric acid (HCl; Merck, New Jersey, US) and heated to 99 °C for 30 min. By treating bacteria with acid and heat, DNA and proteins are degraded, generating LA particles. We pelleted LA particles and washed three times using high speed refrigerated centrifuge (7780; KUBOTA, Osaka, Japan) at 6000 × g for 20 min at 5 °C in PBS (pH 7) before resuspending in PBS at 20 °C. All PBS buffer was diluted from purchased 10x stock PBS (Protech). We counted the particles using a cell counter. We defined the LA particles as 2.5 × 10^9^ particles per unit (1 U).

### 2.2. High-Pressured Homogenization Treatment

We performed a high-pressured homogenization (HPH) treatment for LA particles by Nanolyzer N2 (Gogene Corporation, Hsinchu County, Taiwan). We homogenized the LA particles resuspended in PBS at 25,000 psi by Nanolyzer with 6 passes as LA-C. We homogenized the LA particles resuspended in PBS with addition of trehalose (5%) and emulsifiers (0.1% Tween 20 and 1% Span 80) at 25,000 psi by Nanolyzer N2 with 6 passes as LA-H. 

### 2.3. Dynamic Light Scattering (DLS)

This method measures the intensity fluctuations of scattered light to determine the diffusion coefficient and, ultimately, the hydrodynamic sizes of the particles in the solution. The samples are diluted by 200 times using PBS. The trial was conducted at 25 °C with triplicate. We obtained a Z-average diameter with polydispersity index (PDI) and a size distribution, using Zetasizer Nano S (Malvern Instruments, Worcestershire, UK). 

### 2.4. Mice Immunization Test

Animal experiments were approved by the Committee for Animal Experimentation of National Pingtung University of Science and Technology. We used female *Balb/c* mice (6–8 weeks) purchased from National Laboratory Animal Center, Taiwan, for the mice immunization experiments. We divided the mice into 8 groups of 6 mice each. We immunized all 7 tested groups except the control group through intramuscular injection (IM) with prime vaccination on week 0, followed by one booster vaccination on week 2. We immunized 7 experimental groups with 100 μL of OVA (50 μg of ovalbumin; EndoFit Ovalbumin, Invivogen, San Diego, CA, USA) mixed with different adjuvants: FIA (Freud’s incomplete adjuvant; Sigma, Burlington, USA), Titermax Classic Adjuvant (H4397; Sigma, Burlington, USA), LA-C1 (0.1 U), LA-H1(0.1 U with emulsifiers and trehalose), LA-C5(0.5 U), and LA-H5(0.5 U with emulsifiers and trehalose) and filled to designated volume using PBS. We admixed OVA and different adjuvants or LA just before the immunization. Each mouse received 100 μL of vaccine mixture equally divided across both calf muscles. We collected serum and splenocytes 2 weeks after secondary vaccination from mice and sacrificed to measure OVA-specific IgG.

### 2.5. OVA-Specific IgG Measurement

We detected OVA-specific IgG in sera by an indirect ELISA. Briefly, we coated microtiter plate wells (Nunc) with 100 µL OVA solution (2 μg/well; EndoFit Ovalbumin, Invivogen, San Diego, CA, USA) in 0.2 M carbonate buffer for 24 h at 4 °C. We washed the wells three times with PBS containing 0.05% (*v*/*v*) Tween 20 (PBST) and then blocked with 5% FCS/PBS (AAJ63160AK; Thermo, San Diego, CA, USA) at 37 °C for two hours. After three washings of PBST, 100 μL of a series of diluted sera samples (initial dilution 1:50) and 0.5% FCS/PBS as a control were added to the wells in triplicate. Then, we incubated the plates for 2 h at 37 °C and washed three times with PBST. We diluted from 100 μL aliquot of rabbit anti-mouse IgG horseradish peroxidase conjugate (AP160P; Sigma) with 1:10,000. Then, we incubated the plates for 2 h at 37 °C. After washing with PBST, we determined the peroxidase activity by adding 100 µL of substrate solution (10 mg of O-phenylenediamine and 37.5 µL of 30% H_2_O_2_ in 25 mL of 0.1 M citrate–phosphate buffer, pH 5.0) to each well. After incubation for 10 min at 37 °C, we added 50 µL/well of 2 N H_2_SO_4_ (Merck, NJ, USA) to each well to terminate the enzyme reaction. We used an ELISA reader (TECAN Sunrise^TM^, Männedorf, Switzerland) at 450 nm to perform the assays on each of the sera samples after conducting within- and between-group comparisons.

### 2.6. Cytokine Determination in the Cultured Supernatants of Splenocytes by ELISA

Two weeks after booster vaccination, splenocytes were harvested from vaccinated mice and restimulated with OVA (50 μg/mL; EndoFit Ovalbumin, Invivogen, San Diego, CA, USA). The cultured cells (5 × 10^5^ cells/well) were incubated in complete RPMI 1640 (Gibco) at 37 °C in 5% CO_2_ with ConA (5 g/mL). Following 48 h of culture, we centrifuged the supernatants (prestored at −70 °C) at 1400 × g for 5 min and then determined the levels of INF-γ, IL-2, and IL-10 using mouse ELISA kits (Rapidbio Labs, West Hills, CA, USA). Cytokine level of obtained from the value of OD570. 

### 2.7. Inactivated NDV Antigen and Chicken Vaccination Test

We propagated ND virus (NDV) strain Sato (isolated from National Pingtung University of Technology) in 9-days-old specific pathogen-free (SPF) embryonated chicken eggs that were titrated (EID_50_ = 10^9^) and inactivated by 0.1% formalin at 37 °C for 24 h. We used 4-week-old SPF female chickens and divided into groups of 6 chickens. All chickens received PBS or vaccine (filled to designated volume using PBS) by IM at the breast. In the first chicken vaccination experiment, chickens received 0.5 mL of PBS as the controls, and the experimental-group chickens received inactivated NDV antigen mixed with LA-C5 or LA-H5, and Montanide ISA70 (water-in-oil type; SEPPIC, Courbevoie, France) adjuvants. After two weeks, we provided the booster injection. In the second chicken immunization experiment, chickens of the experimental groups received inactivated NDV antigen mixed with C (carbomers of polyacrylic acid polymer, 1%), O (oil-in-water type with 15% mineral oil adjuvant), LA-H5-C, LA-H5-O, and ISA70. We provided booster shot at 2 weeks after primary injection. During the experiment, we bled all chickens from wing veins. We collected sera at 2-, 4-, 6-, 8- and 10-weeks post-vaccination (first vaccination) and stored at −20 °C until used. We conducted testing for hemagglutination inhibition (HI) in accordance with the Beta method [19].

### 2.8. Mice Acute Toxicity Experiments

We performed acute toxicity test for veterinary vaccines composed of adjuvant followed by the method of Oda et al. (2006). We intraperitoneally injected 10 5-week-old male ICR mice from each group with 0.3 mL/dose of each vaccine. After injection, we weighed the mice every day for seven days. Rate of weight recovery is a measure of mice whose weight recovers to over their weights after injection. We calculated a percentage of body weight change according to the mice weights recorded each day.

### 2.9. Statistical Analysis

Data are expressed as mean ± standard error of the mean (SEM). Statistical differences between groups were determined by one-way analysis of variance (ANOVA) followed by Tukey’s multiple comparison test with IBM SPSS Statistics 20.0 (IBM; Armonk, US) for analysis. Differences regarded as statistically significant between the groups are presented, *: *p* < 0.05, **: *p* < 0.01, ***: *p* < 0.001, and ****: *p* < 0.0001. All graphs were drawn using Graphpad Prism version 8.02 (California, CA, USA). 

## 3. Result

### 3.1. Particle Size 

We performed the particle size distribution assay on the formulated *L acidophilus* particles (LAPs) of different treatments. LA represents the original particle with acidic and high sodium chloride wash, which has a Z-average diameter of 914.27 nm and a polymer dispersity index (PDI) of 0.56 (Figure 1). We created LA-C from LA with high-pressured homogenization (HPH), and LA-H was created by LA in the presence of trehalose and emulsifiers treated with HPH. The average particle sizes of LA-C and LA-H are 510.03 nm and 179.43 nm, respectively (Table 1). The PDIs of LA-C and LA-H were 0.28 and 0.16, respectively, representing their low polydispersity with very uniform particle sizes after HPH treatment. When we added LA-C or LA-H into the solution, no precipitation occurred after being stored at 4 °C and 37 °C for a month. 

### 3.2. Murine Immunization

We tested the immunogenicity of LAP as an adjuvant by vaccinating mice with 50 μg of OVA with LAPs (Figure 2A). We did not vaccinate controls. We vaccinated the experimental groups with a first dose at week 0 and a booster shot at week 2. We collected sera on week 0, 2, and 4 to measure the induced serum antibodies. Mice vaccinated with OVA without any adjuvant induced a poor antibody response. In contrast, robust antibody responses were induced by FIA, Titermax, LA-C1, LA-H1, LA-H1, LA-C5, and LA-H5 at weeks 2 and 4. LA-C5, LA-H5, FIA, and Titermax showed the highest antibody titers at weeks 2 and week 4. However, we observed no significant difference between the four groups. 

We detected cytokine levels from the splenocytes of the vaccinated groups incubated in complete RPMI 1640 for 48 h (Figure 2B). Groups from FIA and Titermax induced a higher IFNγ, IL-2, and IL-10 responses compared to groups of different LAPs. 

### 3.3. Primary Immunization

We vaccinated 4-week-old SPF chickens with inactivated NDV along with different adjuvants. We administered a booster shot two weeks after the first vaccination. We observed a decrease in HI antibody titers in both groups of LA-C5 and LA-H5 on the sixth week post-vaccination compared with the ISA70 group (Figure 3A). These LAPs are aqueous-based solution which may result in a faster drop of antibody titer compared to ISA70, which is mineral-oil-based W/O emulsion. 

### 3.4. Secondary Immunization 

In an attempt to extend the antibody titer formation period with LAPs as immune enhancers in conventional adjuvant applications, we conducted different adjuvant formulations using LAPs as base. We added LAPs in carbomer (C) polymer and O/W emulsions (O). We immunized 4-week-old SPF chickens with inactivated NDV in combination with different adjuvant formulations. We collected serum samples for HI assay to observe the antibody titer (Figure 3B). LA-H5-O adjuvant group exhibited the highest antibody titers at week 4 and maintained a high response up to week 10. LA-H5-C induced the same level of HI titers as LA-H5-O at week 4 but dropped faster at week 6. ISA70 achieved a comparative antibody response through to week 10. Therefore, with the addition of mineral oil such in LAPs achieved a more enduring antibody response. 

### 3.5. Acute Toxicity 

We performed an acute toxicity test regarding LAP adjuvants on the vaccinated groups, using commercial ISA70 as a comparison (Figure 4). After 7 days of recording body weights, ISA70 and LA-H5-O showed significant weight loss compared to PBS and no injection control group. Only LA-H5-C exhibited no significant body weight change when compared with control or PBS groups. 

## 4. Discussion

We evaluated LAPs of different treatments as adjuvants in veterinary vaccine. LA-H, treated with high-pressured homogenization (HPH) in addition of trehalose and emulsifiers, exhibited the lowest Z-average diameter of 179.43 nm compared to LA-C, treated with HPH, and LA of basic treatment. Particle size of lower than 200 nm appears to have more advantage over micro-particles (>1 μm) at priming cytotoxic T cells in vaccines [20], suggesting the possibility of higher immune stimulation from LA-H. LA-H also exhibited the lowest PDI value of 0.16 all three LAPs indicating a more evenly distributed particle size. No precipitation was seen after being stored at 4 °C and 37 °C for a month, but more stability conditions are still needed to analyzed. 

LAPs adjuvant efficacy was evaluated in mice with ovalbumin (OVA) as antigens. Five-fold dosage of LA-C and LA-H which were named LA-C5 and LA-H5, respectively, exhibited similar antibody response compared to commercial adjuvants such as IFA and Titermax and better than all one-fold dosage. Comparing the two one-fold dosage, LA-H1 induced a higher antibody response than LA-C1, suggesting that the addition of trehalose and emulsifiers is positively correlated with antibody stimulation and smaller particle sizes may be a key factor for antibody induction. A Th1 response is characterized by the secretion of cytokines, such as IL-2, IFN-γ, IL-12, and GM-CSF, which activate cytotoxic T cells and macrophages in inducing CMI. However, a Th2 response induces cytokines such as IL-4, IL-5, and IL-10, which activate B lymphocytes for the induction of the humoral immune response [21]. Groups from FIA and Titermax induced higher IFNγ, IL-2, and IL-10 responses. Vaccinations with LAPs appeared to induce a Th1 or Th2 response bias compared with vaccinations with OVA and control groups.

Immunization with poultry using inactivated Newcastle disease virus (NDV) as antigens was conducted for evaluation of adjuvants. LA-H5 and LA-C5 ‘s antibody titer value decreased at week 8 to 10 indicating the immune stimulation cannot last till week 8. However, commercial W/O adjuvant ISA70 can maintain a high value of HI through to week 10. ISA70 is a water-in-oil-type (W/O) adjuvant composed of mineral oil and refined emulsifier. When ISA70 is mixed with aqueous antigens, the continuous mineral oil phase can cause a slower antigen release which can last longer than LAPs. Moreover, the aqueous-based LAPs may not act as an effective adsorbent for antigens binding or may not be stable enough to last through to week 8, resulting a lower HI value at week 8 and 10 [22,23].

Formulation of LAPs was adjusted to achieve a more sustainable antibody response. Addition of carbomer or mineral oil was added to LAPs for new formulations in the poultry immunization trial using NDV. We observed that the LA-H5-O adjuvant group exhibited the highest antibody titers at week 4 and remained a high titer value to week 10, which was nearly as high as ISA70. Furthermore, LA-H5-C induced the same level of HI value at week 4 compared to LA-H5-O but declined rapidly at week 6, perhaps because carbomer polymer adsorbed less antigen than oil droplets. In addition, the immune response induced by dispersed mineral oil droplets was stronger than in the aqueous polymers, suggesting LA-H5-O as a possible candidate for poultry vaccine adjuvant against ND or other diseases. 

Acute toxicity was analyzed, and body weight changed was observed in all vaccinated groups, suggesting a possible toxicity in these adjuvants. The adjuvant of LA-H5-C exhibited less body weight change with no significance compared with PBS and control group, indicating a minor toxicity to host.

## 5. Conclusions

*L acidophilus* is a Gram-positive bacteria considered safe for humans and animals, and LAPs made by LA can stimulate immune responses via TLR2 receptors [24,25]. Using high-pressured homogenization (HPH), LAPs alone or formulated with trehalose and emulsifiers have shown considerable scope for stability and possible vaccine application. We proved the adjuvanticity of LAPs in murine models, revealing their potential as adjuvants in vaccine applications. Although LAPs alone could induce HI antibody levels similar to a water-in-oil-type ISA70 adjuvant at weeks 4 and 6, they declined at a faster rate than ISA70 at weeks 8 and 10 in poultry vaccination trials. To extend the duration of immune response, LAPs added with mineral oil or carbomer can last longer in antibody titer value and have showed similar adjuvanticity to ISA70. However, both LA-H5-O and ISA70 exhibited more weight loss compared with LA-H5-C which may be caused by adjuvant toxicity. Therefore, LA-H5-C could be considered as a safter and effective choice of vaccine adjuvant in animals. Adjuvant stability trials in different conditions such as temperature and humidity and efficacy in different animals still need to be examined. Evaluation of a more detailed toxicity trial is needed to be considered as safe for all animals in veterinary use [26]. 

## Figures and Tables

**Figure 1 vetsci-09-00698-f001:**
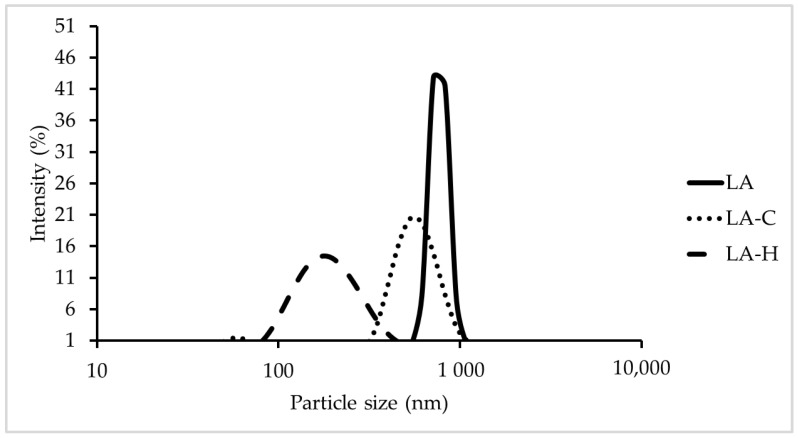
Particle size of processed adjuvants. LA: *L. acidophilus* particles. LA-C: LA particles treated with high-pressurized homogenization (HPH). LA-H: LA particles treated with HPH with addition of trehalose and emulsifiers.

**Figure 2 vetsci-09-00698-f002:**
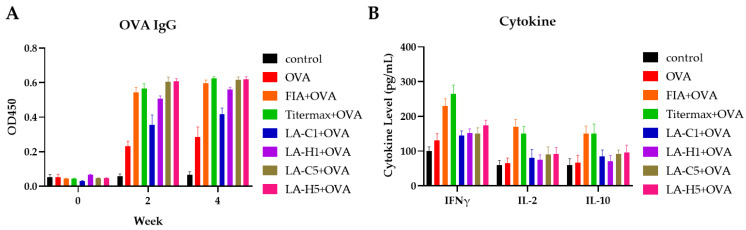
Antibody and cytokine stimulation of vaccinated mice. (**A**) Antibody titer of OVA-induced IgG. *p* > 0.5 (compared with control) (**B**) Cytokine level of IFNγ, IL-2, and IL-10 from vaccinated mice splenocytes incubated for 48 h. *p* > 0.5 (compared with control). Control: no vaccination. OVA: ovalbumin. FIA+OVA: Freud’s incomplete adjuvant with OVA. Titermax+OVA; Titermax adjuvant with OVA. LA-C1(0.1 U), LA-H1(0.1 U), LA-C5(0.5 U), LA-H5(0.5 U). LA-C: LA particles treated with HPH. LA-H: LA particles treated with HPH with addition of trehalose and emulsifiers. Data are presented as the mean ± SEM and determined by one-way ANOVA using Tukey’s posttest.

**Figure 3 vetsci-09-00698-f003:**
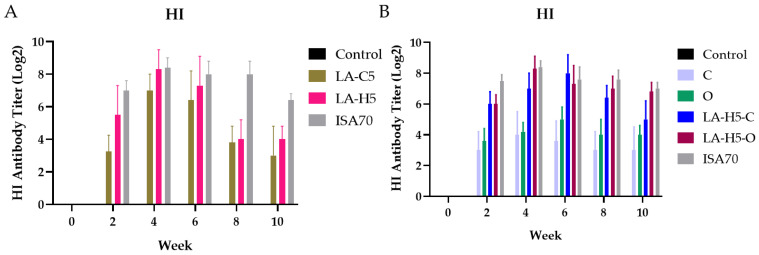
Antibody titer from hemagglutinin inhibition assay (HI) with designated adjuvants of (**A**) primary immunization and (**B**) secondary immunization. Control: no vaccination. LA-C5: 0.5 U LA particles. LA-H5: 0.5 U LA particles with trehalose and emulsifiers. ISA70: Montanide ISA70 W/O adjuvant. LA-H5: *, ISA70: ** (compared with control) in Figure 3A; LA-H5-C: **, LA-H5-O: **, ISA70: ** (compared with control) in Figure 3B. *: *p* < 0.05, **: *p* < 0.01. Data are presented as the mean ± SEM and determined by one-way ANOVA using Tukey’s posttest.

**Figure 4 vetsci-09-00698-f004:**
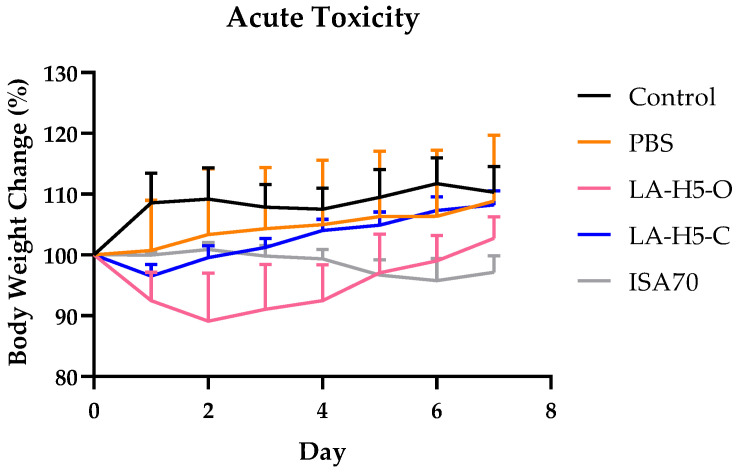
Acute toxicity assay of the vaccines. LA-H5-O: ****, LA-H5-C: *, ISA70: **** (compared with control). LA-H5-O: ***, LA-H5-C: P >0.5, ISA70: * (compared with PBS), determined by one-way ANOVA using Tukey’s posttest. Control: no vaccination. C: 1% carbomer. O: mineral oil with water-in-oil formulation. LA-H5-C: 0.5 U LA particles with trehalose, emulsifiers, and carbomer. LA-H5-O: 0.5 U LA particles with trehalose and emulsifiers, and mineral oil with water-in-oil formulation. ISA70: Montanide ISA70 adjuvant. **: p* < 0.05, ***: p* < 0.01, *****: *p* < 0.001, and ******: *p* < 0.0001. Data are presented as the mean ± SEM and determined by one-way ANOVA using Tukey’s posttest.

**Table 1 vetsci-09-00698-t001:** Z-average diameter and polymer dispersity index (PDI) of adjuvants.

Name	Z-Average Diameter (nm)	PDI
LA	914.27	0.56
LA-C	510.03	0.28
LA-H	179.43	0.16

## Data Availability

The original data generated during this study are included in this published article.

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
