# Peer review of "Particulate Cell Wall Materials of Lactobacillus acidophilus as Vaccine Adjuvant"

_vetsci, 2022, doi:10.3390/vetsci9120698_

Round 1
Reviewer 1 Report
The authors have evaluated nanoparticulate formulations of Lactobacillus acidophilus as vaccine adjuvants, using IgG antibody response to ovalbumin in mice and hemaglutinin titers to Newcastle disease virus in chickens as criteria of efficacy. Several specific issues: There is no mention of the immunogens used in the abstract. The use of the word "respectively" throughout the manuscript is improper and should be removed, except for Section 3.1, where it is used properly. Details of LA culture, such as inoculum size, medium, culture size and method, should be provided. What kind of cell counter was used? What was the OVA concentration in the ELISA coating solution? ELISA wells should not be washed with detergent-containing buffer before blocking. An ELISA should be repeated with no washes prior to blocking to determine whether the results are accurate. In Section 2.5, "liters" are used twice instead of "microliters." What ELISA reader was used? Quantities of immunogen and adjuvant used for chicken immunizations should be specified. The LA particle size specified in Section 3.1 and Table 1 is 1514.66 nm, but the peak position in Figure 1 is less than 1,000 nm. The first dose and the booster shot cannot both be at week 0, unless they were administered less than 7 days apart, in which case the day should be specified. In Section 3.2, it should not be said that there seemed to be no significant difference between the groups. Significance level (e.g., p = 0.05) should be defined and if p > significance level, it should be stated that differences were not significant. There certainly is a significant difference between the adjuvant groups and the 0-point values and the controls; p-values should be specified. In Figure 2A, it looks like the LA-C1+OVA responses may be significantly lower than the other adjuvant groups. In Figure 4B, it seems that LA-H5-O may show some toxicity. The colors used for that and the LA-H5-C line are difficult to distinguish. Another color should be used for one of them.
Author Response
Answer to Reviewer 1: the answers are answered in BLUE printing
The authors have evaluated nanoparticulate formulations of Lactobacillus acidophilus as
vaccine adjuvants, using IgG antibody response to ovalbumin in mice and hemaglutinin titers
to Newcastle disease virus in chickens as criteria of efficacy. Several specific issues: There is
no mention of the immunogens used in the abstract. The use of the word "respectively"
throughout the manuscript is improper and should be removed, except for Section 3.1, where
it is used properly.
A: English editing has been done using MDPI’s English editing service.
Details of LA culture, such as inoculum size, medium, culture size and method, should be
provided. What kind of cell counter was used? What was the OVA concentration in the
ELISA coating solution? ELISA wells should not be washed with detergent-containing
buffer before blocking. An ELISA should be repeated with no washes prior to blocking to
determine whether the results are accurate. In Section 2.5, "liters" are used twice instead of
"microliters." What ELISA reader was used? Quantities of immunogen and adjuvant used
for chicken immunizations should be specified.
A: A much more detailed material and method has been added to the manuscript.
The LA particle size specified in Section 3.1 and Table 1 is 1514.66 nm, but the peak position
in Figure 1 is less than 1,000 nm.
A: There is an error in deciphering the mean value of diameter. This should be more excised
as Z-average diameter and the value is wrong. It should be 914.27.
The first dose and the booster shot cannot both be at week 0, unless they were administered
less than 7 days apart, in which case the day should be specified.
A: An error in typing and have revised to week 2.
In Section 3.2, it should not be said that there seemed to be no significant difference between
the groups. Significance level (e.g., p = 0.05) should be defined and if p > significance level, it
should be stated that differences were not significant. There certainly is a significant
difference between the adjuvant groups and the 0-point values and the controls; p-values
should be specified.
A: Significance regarding the groups has been added to the figures.
In Figure 2A, it looks like the LA-C1+OVA responses may be significantly lower than the
other adjuvant groups. In Figure 4B, it seems that LA-H5-O may show some toxicity. The
colors used for that and the LA-H5-C line are difficult to distinguish. Another color should
be used for one of them.
A: One-way annova and multiple comparison (Tukey) has been conducted and showed
significant indicating possible toxicity to the adjuvants. Colors has been changed to the graph
to make it clearer.
Reviewer 2 Report
Summary:
The article “Particulate Cell Wall Materials of Lactobacillus acidophilus as Vaccine Adjuvant” presents the use of particles of L. acidophilus that have been created using high pressure and chemical treatment. These particles were then use to as adjuvants in 1) mice vaccinated with ovalbumin and 2) chickens vaccinated with formalin killed Newcastle virus. Immunogenicity was assessed in mice as ovalbumin-IgG and via hemagglutinin inhibition assay for the chickens. Additionally, cytokines from splenocytes were measured by ELISA and acute toxicity in mice was assessed by IP injection and then measurement of body weight change.
General concept comments:
Strengths: This was an interesting article assessing particles from Lactobacillus acidophilus as a vaccine adjuvant. New adjuvant strategies are needed for vaccines and this work assesses a novel new adjuvant which was delivered with both protein (ovalbumin) and an inactivated virus. Important assessments of response to vaccination were measured including antibody induction and acute toxicity in mice.
Weaknesses: The work here is important and the flow of the manuscript is generally acceptable. The manuscript does suffer from flaws that require major editing prior to recommendation for publication. The most striking issues involve the materials and methods and a lack of statistics. General and specific comments are listed below.
1. Minor English editing needed though out.
2. Materials and methods need major revisions (please see specific comments). Generally, methods do not allow for reproducibility and manufacturers of reagents are not listed.
3. No methods for statistics are listed.
4. Conclusion is exceedingly brief and must be expanded.
Specific comments:
Introduction:
1. Lines 51-54: The authors state that Th1 cells induce cell mediated immunity which increases B cell antibody secretion. Cell mediated immunity generally refers to cytotoxic T cells and destruction of intracellular pathogens. This sentence highlights antibody secretion which would be more inline with humoral or antibody mediated immunity. This needs to be rewritten to make it clear which time of immunity the author is discussing.
2. Line 59: remove “a kind of”
3. Lines 64-66: what is the mechanism of action of the enhancer matrix (GEM); please give more background on GEM as the process and mechanism of action is important to this manuscript
4. Lines 69-77: please state how the adjuvant is delivered: oral vs intramuscular vs intra-peritoneal etc.
Materials and methods:
1. Lines 80-85:
a. What is the strain of Lactobacillus acidophilus?
i. How was it obtained?
b. Manufacturer of HCl, PBS?
c. How much culture or CFU of bacteria were suspended in HCl?
d. Speed at which LA were pelleted?
e. Line 85: The definition of an LA particle as a “non-living particle” is confusing to this reader. Please revise.
2. Lines 93-96: This section defines what dynamic light scattering is but not how it was performed.
3. Lines 97-109:
a. What is the sex of the animals used in these groups?
b. How was the Ova obtained?
c. What type of buffer was used to resuspend the Ova and adjuvants?
d. How were the animals euthanized?
e. How was the spleen processed for splenocytes?
4. Lines 110-123:
a. How was the Ova solution obtained?
b. What is the concentration of the Ova solution for plate coating?
c. What buffer was used for coating?
d. What was the dilution buffer for the samples?
e. How was it determined what the titer is? What is the within-group and between-group comparison?
5. Lines 125-128:
a. How were the splenocytes collected/processed?
b. Media for cell culture?
6. Lines 130-144:
a. How was the virus “titrated”?
b. Where was the ND virus originally obtained?
c. What was the buffer used for vaccination?
d. Sex of the chickens?
e. What was the concentration of the adjuvants used?
Results:
1. Line 163: what are the different temperatures?
2. Line 162: How was precipitation determined?
3. Line 172: States that first dose was delivered at week 0 and booster also at week 0?
4. Line 177: States no significant difference but how the statistics were performed is not listed.
5. Lines 187-189: No statistics are available to assess if there are differences between the cytokine levels without statistics the conclusion made by the authors of a Th1 v Th2 response is premature.
6. Lines 199-200: Please define “sharper decrease”.
7. Lines 198-201: No statistics so interpretation by the authors is premature.
8. Lines 202-205: This interpretation should be moved to the conclusions section.
9. Lines 217-219: No statistics so interpretation of these results is premature.
Figures:
1. Figure 1: What does intensity on the Y-axis refer to?
2. Figure 2:
a. This reader would like to see the individual mouse values on the graph and not just the average OD450.
b. Line 191: The use of “titer” is incorrect as titer was not determined in this test as the results appear to just be one dilution OD.
3. Figure 3: Similar to figure 2, this reader would like to see the individual HI values and not just the averages.
4. Figure 4 A and B: These figures are not related. The reader suggests combining 4A with figure 3.
Conclusions:
1. Line 235: states that lactobacillus is a gram-negative bacteria. This is incorrect it is gram positive.
2. Lines 235-237: LA does not stimulate TLR4. The references cited are incorrectly cited and should be reviewed by the authors. Reference 22 states that Porphyromonas gingivalis stimulates TLR4 and reference 23 states that there is increased expression of TLR4 on cells following exposure to Lactobacillus casei.
3. Can the authors please comment on next steps and future directions?
Author Response
Answer to Reviewer 2: the answers are answered in BLUE printing
- Minor English editing needed though out.
A: English editing has been conducted using MDPI’s English editing service
- Materials and methods need major revisions (please see specific
comments). Generally, methods do not allow for reproducibility and
manufacturers of reagents are not listed.
A: Specific methods including the manufacturers of used buffers have added
to the manuscript.
- No methods for statistics are listed.
A: Statistics has been added to the last part of material and methods.
- Conclusion is exceedingly brief and must be expanded.
A: Conclusion has expended in including future work.
Specific comments:
Introduction:
- Lines 51-54: The authors state that Th1 cells induce cell mediated
immunity which increases B cell antibody secretion. Cell mediated
immunity generally refers to cytotoxic T cells and destruction of
intracellular pathogens. This sentence highlights antibody secretion
which would be more inline with humoral or antibody mediated
immunity. This needs to be rewritten to make it clear which time of
immunity the author is discussing.
A: The paragraph has been adjusted.
- Line 59: remove “a kind of”
A: Removed
- Lines 64-66: what is the mechanism of action of the enhancer matrix
(GEM); please give more background on GEM as the process and
mechanism of action is important to this manuscript
A: A detailed description is added.
- Lines 69-77: please state how the adjuvant is delivered: oral vs
intramuscular vs intra-peritoneal etc.
A: “intramuscular” is added to the paragraphs.
Materials and methods:
- Lines 80-85:
- What is the strain of Lactobacillus acidophilus?
- How was it obtained?
A: LA (BCRC 17049) was bought from Bioresource Collection and Research
Center, Hsinchu City, Taiwan.
- Manufacturer of HCl, PBS?
A: Added to the manuscript.
- How much culture or CFU of bacteria were suspended in HCl?
A: 10L of culture bacteria was centrifuged and treated in HCl
- Speed at which LA were pelleted?
A: 6000xg using High Speed Refrigerated Centrifuge (7780; KUBOTA) for 20
minutes.
- Line 85: The definition of an LA particle as a “non-living particle” is
confusing to this reader. Please revise.
A: It is revised to “ particle”.
- Lines 93-96: This section defines what dynamic light scattering is but not
how it was performed.
A: Detailed information is added.
- Lines 97-109:
- What is the sex of the animals used in these groups?
A: female
- How was the Ova obtained?
A: purchased and added to the manuscript.
- What type of buffer was used to resuspend the Ova and adjuvants?
A: PBS (added to the manuscript)
- How were the animals euthanized?
A: the animals are anesthetized.
- How was the spleen processed for splenocytes?
A: added to the manuscript
- Lines 110-123:
- How was the Ova solution obtained?
A: OVA was bought from INVIVOGEN.
- What is the concentration of the Ova solution for plate coating?
A: 2 μg/well
- What buffer was used for coating?
A: 0.2 M carbonate buffer.
- What was the dilution buffer for the samples?
A: Dilute with carbonate buffer
- How was it determined what the titer is? What is the within-group and
between-group comparison?
A: The answers to these questions has been added to the manuscript.
- Lines 125-128:
- How were the splenocytes collected/processed?
A: Detailed process has been added to the manuscript.
- Media for cell culture?
A: RPMI 1640 (added to the manuscript)
- Lines 130-144:
- How was the virus “titrated”?
A: with serial dilution of 10x.
- Where was the ND virus originally obtained?
A: origin added to the manuscript (from National Pingtung University of
Technology).
- What was the buffer used for vaccination?
A: PBS
- Sex of the chickens?
A: Female
- What was the concentration of the adjuvants used?
Results:
- Line 163: what are the different temperatures?
A: at 4 °C and 37 °C
- Line 162: How was precipitation determined?
A: Precipitation was determined from appearance in 15-mL tube of possible
different layers. Photos were taken for a month in every three days to keep
track of the appearance of the vaccine.
- Line 172: States that first dose was delivered at week 0 and booster also at
week 0?
A: An error to writing. It should be week 2 for the booster shot.
- Line 177: States no significant difference but how the statistics were
performed is not listed.
A: Statistics has been added to the figure.
- Lines 187-189: No statistics are available to assess if there are differences
between the cytokine levels without statistics the conclusion made by the
authors of a Th1 v Th2 response is premature.
A: Statistics has been added to the figure.
- Lines 199-200: Please define “sharper decrease”.
A: Revised to “decrease”.
- Lines 198-201: No statistics so interpretation by the authors is premature.
A: Statistics has been added to the figure.
- Lines 202-205: This interpretation should be moved to the conclusions
section.
A: A brief description of the result of vaccination.
- Lines 217-219: No statistics so interpretation of these results is premature.
A: Statistics has been added to the figures and revised the methods as well.
Figures:
- Figure 1: What does intensity on the Y-axis refer to?
A: intensity refers to the signal at OD570.
- Figure 2:
- This reader would like to see the individual mouse values on the graph
and not just the average OD450.
- Line 191: The use of “titer” is incorrect as titer was not determined in this
test as the results appear to just be one dilution OD.
- Figure 3: Similar to figure 2, this reader would like to see the individual
HI values and not just the averages.
- Figure 4 A and B: These figures are not related. The reader suggests
combining 4A with figure 3.
A: 4A has combined to figure 3. Individual values would result in in more
confused graph.
Conclusions:
- Line 235: states that lactobacillus is a gram-negative bacteria. This is
incorrect it is gram positive.
A: Revised
- Lines 235-237: LA does not stimulate TLR4. The references cited are
incorrectly cited and should be reviewed by the authors. Reference 22
states that Porphyromonas gingivalis stimulates TLR4 and reference 23
states that there is increased expression of TLR4 on cells following
exposure to Lactobacillus casei.
A: Reference revised and deleted.
- Can the authors please comment on next steps and future directions?
A: Future has been added in regarding adjuvant stability in temperature and
humidity, efficacy in different animals, and toxicity.
Round 2
Reviewer 1 Report
There is still no mention of the immunogens used (OVA and NDV) in the abstract. Remove "respectively" from pg. 2, line 75. In Figure 2, it would be best to leave out mention of P-values, since the relative response levels are evident. On pg. 7, line 258, remove B from Figure 4B, since there is now only one Figure 4.
Author Response
There is still no mention of the immunogens used (OVA and NDV) in the abstract.
A: OVA and NDV added in the abstract.
Remove "respectively" from pg. 2, line 75.
A: Removed
In Figure 2, it would be best to leave out mention of P-values, since the relative response levels are evident.
A: Removed
On pg. 7, line 258, remove B from Figure 4B, since there is now only one Figure 4.
A: Removed.
Reviewer 2 Report
I thank the authors for the corrections and responses. The manuscript is much improved and in its current form needs only minor changes for publication. Please see specific comments below:
Summary:
- Line 19-20: “We compared ISA70 … - this sentence is confusing, please revise as it is unclear which treatment group had a higher titer or a titer that decreased more quickly.
Abstract:
- Line 33: “in” should be “as a”
Introduction:
- Line 57: “between” should be “of”
Methods:
- The reviewer asked for concentrations of the adjuvants used but this was not addressed in the responses. Some concentrations were added to the methods but not all. Please include.
Results:
- Lines 213-214: “splenocytes were incubated for 48 hours” – in what were they incubated in?
- Lines 216-219: no data is shows of the IL-4, IL-5, IL-10 cytokine – should include this data in Figure 2 or as supplemental
- line 233 – ISA70 and ISA 70. Throughout the manuscript this is not consistently written, and the author should check and correct.
- Lines 236-237- Is there evidence that the LAPs entrap antigen? The reference used here is in reference to the ISA70 and stating that the LAPs “entrapped” antigens may not be an accurate statement. Please cite a paper or include data that shows entrapment of antigen or remove.
- Line 251 – again can the authors cite reference or show results that the “polymer absorbed” the antigen (see above)
- Line 258 – no figure 4B
- Line 259 – “showed significant” – please state the p value
Conclusions:
- Line 280 – please define toxicity as weight loss since this is the only measure that was evaluated in this study
Figures:
- Figure 2:
o P value is listed in the figure caption but it is not indicated on the figure or in the caption which group comparison this refers to. Please indicate in the figure or caption.
- Figure 3:
o Please restate what * and ** mean in the figure caption.
o Do the * and ** refer to graph A or B?
o Is this LA-H5-C or -O?
Author Response
I thank the authors for the corrections and responses. The manuscript is much improved and in its current form needs only minor changes for publication. Please see specific comments below:
Summary:
- Line 19-20: “We compared ISA70 … - this sentence is confusing, please revise as it is unclear which treatment group had a higher titer or a titer that decreased more quickly.
A: Sentence was revised.
Abstract:
- Line 33: “in” should be “as a”
A: Corrected.
Introduction:
- Line 57: “between” should be “of”
A: Corrected
Methods:
- The reviewer asked for concentrations of the adjuvants used but this was not addressed in the responses. Some concentrations were added to the methods but not all. Please include.
Results:
- Lines 213-214: “splenocytes were incubated for 48 hours” – in what were they incubated in?
A: They were incubated in complete RPMI 1640.
- Lines 216-219: no data is shows of the IL-4, IL-5, IL-10 cytokine – should include this data in Figure 2 or as supplemental
A: In Figure 2, we have the data of IL-10, but we did not do IL-4 and IL-5.
- line 233 – ISA70 and ISA 70. Throughout the manuscript this is not consistently written, and the author should check and correct.
A: Checked
- Lines 236-237- Is there evidence that the LAPs entrap antigen? The reference used here is in reference to the ISA70 and stating that the LAPs “entrapped” antigens may not be an accurate statement. Please cite a paper or include data that shows entrapment of antigen or remove.
A: The word “entrap” is not correct. It should be “adsorb”. The antigens will absorb onto LAPs with acid wash similar with Gram-positive-enhancer-matrix(GEM)-related articles.
- Line 251 – again can the authors cite reference or show results that the “polymer absorbed” the antigen (see above)
A: Reference cited.
- Line 258 – no figure 4B
A: Corrected.
- Line 259 – “showed significant” – please state the p value
A: P value has added to figure caption.
Conclusions:
- Line 280 – please define toxicity as weight loss since this is the only measure that was evaluated in this study
A: Adjuvant toxicity may be one of the reasons for weight loss. More evaluation may be needed to define toxicity.
Figures:
- Figure 2:
o P value is listed in the figure caption but it is not indicated on the figure or in the caption which group comparison this refers to. Please indicate in the figure or caption.
A: Comparison details is added in the figure caption.
- Figure 3:
o Please restate what * and ** mean in the figure caption.
o Do the * and ** refer to graph A or B?
o Is this LA-H5-C or -O?
A: Figure caption has been revised.